# Support Vector Machine as a Supervised Learning for the Prioritization of Novel Potential SARS-CoV-2 Main Protease Inhibitors

**DOI:** 10.3390/ijms22147714

**Published:** 2021-07-19

**Authors:** Nedra Mekni, Claudia Coronnello, Thierry Langer, Maria De Rosa, Ugo Perricone

**Affiliations:** 1Department of Pharmaceutical Chemistry, University of Vienna, 1090 Vienna, Austria; thierry.langer@univie.ac.at; 2Drug Discovery Unit, Fondazione Ri.MED, 90128 Palermo, Italy; ccoronnello@fondazionerimed.com (C.C.); mderosa@fondazionerimed.com (M.D.R.)

**Keywords:** machine learning, classification, main protease, COVID-19, molecular docking

## Abstract

In the last year, the COVID-19 pandemic has highly affected the lifestyle of the world population, encouraging the scientific community towards a great effort on studying the infection molecular mechanisms. Several vaccine formulations are nowadays available and helping to reach immunity. Nevertheless, there is a growing interest towards the development of novel anti-covid drugs. In this scenario, the main protease (Mpro) represents an appealing target, being the enzyme responsible for the cleavage of polypeptides during the viral genome transcription. With the aim of sharing new insights for the design of novel Mpro inhibitors, our research group developed a machine learning approach using the support vector machine (SVM) classification. Starting from a dataset of two million commercially available compounds, the model was able to classify two hundred novel chemo-types as potentially active against the viral protease. The compounds labelled as actives by SVM were next evaluated through consensus docking studies on two PDB structures and their binding mode was compared to well-known protease inhibitors. The best five compounds selected by consensus docking were then submitted to molecular dynamics to deepen binding interactions stability. Of note, the compounds selected via SVM retrieved all the most important interactions known in the literature.

## 1. Introduction

The COVID-19 pandemic, also known as Severe Acute Respiratory Syndrome Coronavirus-2 (SARS-CoV-2) is afflicting the health and routines of billions of people worldwide.

During the last few months, we are witnessing a race against time to vaccinate as many people as possible; however, the disparities in vaccine distribution between countries and the new emerging variants represent a further public health concern, making it hard to reach a full immunization [1,2].

SARS-CoV-2 is a member of the betacoronavirus family, together with SARS-CoV and Middle East Respiratory Syndrome (MERS-CoV). The enormous scientific effort worldwide led to a better understanding of SARS-CoV-2 structure and the infection mechanism, spotting four main druggable targets, namely the Spike (S) protein, Papain-like protease (PLpro), RNA-dependent RNA polymerase (RdRp) and the main protease/3C-like protease (Mpro/3CLpro) [3,4]. In particular, SARS-CoV-2 Mpro leads a crucial role in the viral replication process. Mpro is a cysteine protease responsible for the cleavage of polypeptides during the viral genome transcription, promoting the generation of non-structural proteins, which can assemble to form new infectious virions. As shown in Figure 1, the Mpro catalytic site includes four subsites, namely S1, S2, S3 and S4, hosting the binding site of protease inhibitors. [5]. Of special importance, the catalytic dyad is enclosed into the S1/S2 pockets and includes the Cys 145 and His 41 residues. According to the inhibition mechanism, His 41 activates the thiol group of Cys 145 (SH), which, in turn, performs the nucleophilic attack to the substrate. Gln 189, near S3, confers plasticity to the pocket, while Glu 166 (S4) is involved in the connection between the dimer interface and the substrate biding site, thus having a key role for the catalytic activity [6,7].

Inhibition of the Mpro enzyme blocks the SARS-CoV-2 life cycle; for this purpose, Mpro represents an appealing target for the development of new potential inhibitors.

There is an urgent need to discover new drugs to help fight the global pandemic.

In this scenario, in silico virtual screening (VS), provides a cost-effective and a more rapid approach for lead compounds discovery, especially when compared to the traditional high-throughput screening (HTS) process.

However, vs. has some limitations, such as the inaccuracy of scoring functions, the partial account of ligands flexibility and the receptor plasticity [8]. Altogether, these factors could lead to a low hit rate and a low enrichment factor [9].

In the last two decades, the machine learning (ML) approach has been explored in the field of drug discovery, showing an ever-growing success and overcoming vs. drawbacks.

In this study, we exploited ML techniques to develop a support vector machine (SVM) model in order to identify potential novel Mpro inhibitors, as a prior classification step before performing a structure-based prospective vs. on the Mpro protein.

PostEra start-up, in collaboration with Diamonds, launched a crowdsourced initiative in order to boost the discovery of new antiviral compounds against SARS-CoV-2 Mpro [10,11]. The main goal was to design and biologically evaluate as many inhibitors as possible, in order to rapidly develop new therapeutics. This initiative, namely, COVID Moonshot, offers a platform collecting molecules designed by several research groups around the world. PostEra COVID-19 activity data are indeed an interesting data source reporting a collection of compounds with known inhibitory activities against Mpro.

In our study, the PostEra COVID-19 Moonshot dataset was used as a data source for the development of a supervised classification model able to discriminate the activity against Mpro from a pool of unseen compounds. More specifically, our classification model was trained using 1D and 2D molecular descriptors calculated for the COVID Moonshot compounds and the inhibitory activities against Mpro were set as a label.

In order to get a reliable classification model, the main focus was the feature selection protocol prior to modelling. This workflow task allowed the selection of the most relevant molecular descriptors able to correlate compound chemical structures to their activity against Mpro. In this regard, feature selection is a challenging task, as it should be able to detect a relationship between molecular descriptors and biological activity, starting from a group of descriptors. A too high number of descriptors, compared to the observations, could negatively affect the analysis, bringing to a misleading association between the features and the bioactivity, due to an overfitting error.

The selection of a descriptors subset strongly correlated to the biological activity contributes to a higher model learning efficiency and improves the performance of the classification model. Simultaneously, the computational complexity is reduced thanks to a decreased number of features [12].

In this study, a random forest approach, combined with recursive feature elimination with cross validation (RF–RFE–CV) [13,14,15], was performed for the feature selection in order to achieve good performance with moderate computational efforts. Through the application of a feature selection protocol, we explored the ability of our model to eliminate irrelevant features, to reduce data dimensionality and to lead to the recruitment of the most informative molecular descriptors. Selected molecular descriptors were then used for the development of the SVM model for the classification of new SARS-CoV-2 Mpro inhibitors. In parallel, structure-based approaches were used to explore the main protein–ligand interactions and their stability. Docking protocol was validated and the compounds predicted as active by SVM were submitted to docking and molecular dynamics. The evaluation of the binding mode allowed us to identify the most promising putative Mpro inhibitors.

## 2. Results and Discussion

### 2.1. Feature Selection with the RF–RFE–CV Method

Feature selection was performed through the implementation of a python3 script using Sklearn libraries. The script is available at GitHub repository [16].

Feature selection was carried out on the training set, in order to identify the crucial molecular descriptors able to explain the possible correlation between Mpro inhibitors activity and their chemical structures. Particularly, random forest recursive feature elimination (RF–RFE) was implemented in order to select relevant molecular descriptors [17,18]. According to the RF–RFE procedure, each feature was weighed, evaluated and recursively eliminated if not relevant. The process stopped when the most important features were identified and no further features needed to be eliminated to maintain the performance of the whole prediction model. The outcome of RF–RFE was recursively validated with k-fold cross validation (CV), leading to the automatic tuning of the number of features to be selected and to define the optimal number of decisional trees to build the forest.

The feature selection process is related to the number of trees populating the forest and to the correlation threshold set for molecular descriptors. Highly correlated variables do not add any further information. It is worth mentioning here that the number of trees was not known a priori and it was crucial to set it in order to obtain an accurate model. Aiming at finding the optimal number of trees and the best correlation threshold, we performed feature selection by using 1, 10, 100 and 500 trees. For each RF, the descriptors correlation threshold was set in a range of 0.60–0.90 using a step size of 0.1. The selected descriptors were analysed from feature selection based on the best results (see SVM development and evaluation shown in Table 4). By using RF–RFE–CV, the total number of descriptors was reduced. Table 1 shows the selected molecular descriptors distribution though the RF–RFE–CV method along with the descriptor type and number of trees. The largest number of selected descriptors belonged to autocorrelation and atom type electro-topological state (E-State) families. According to the literature [19,20], these two descriptor classes are known to be prominent for the identification of proteases inhibitors, as they refer to the electronic contour of structures. For the covalent inhibition, the electronic and polarizability characteristic of the reacting moiety (aldehydes, α-keto-esters, nitriles, etc.) is crucial for the reaction to happen. For non-covalent inhibitors, the molecular surface electronic features are equally important, due to the H-bond and π-π network stabilizing the ligand within the protein catalytic site [21].

Autocorrelation descriptors encode the molecular structure and the physicochemical properties assigned to the atoms [12]. E-State values encode the information concerning the electron accessibility at the atom level. In this regard, the E-state index expresses the potentially noncovalent intermolecular interactions [13].

Each of the four lists of features, selected by changing the number of trees in the RF–RFE–CV pipeline, was used to train an SVM, as described in the next paragraph.

### 2.2. SVM Training and Testing

The SVM purpose is to find the best separating hyperplane, able to maximize the margin between the two classes (e.g., active–inactive) [22].

Hyperparameters, such as the kernel type, C and gamma type, were tuned and mainly contributed to the model performance [23,24]. In detail, we implemented a grid searching algorithm able to consider and evaluate all hyperparameter combinations with a cross validation approach. In Table 2, the best SVM hyperparameters found, when 1, 10, 100 and 500 trees were set for the random forest method, are reported. Each SVM model was trained by using the selected features summarized in Table 1.

Our classification model was evaluated as a function of descriptors correlation threshold and the number of decisional trees.

Depending on these parameters, we observed different accuracy and precision values. In particular, accuracy indicates the fraction of correct predictions from our model, while precision quantifies the fraction of correctly predicted positive observations. Table 3 reports our models performance evaluators.

The best precision and accuracy values were obtained when 100 trees were set, excluding features with a correlation higher than 0.75. The seven features used to train the best model are listed in Table 4.

According to these results, we identified the most relevant molecular descriptors explaining the relation between molecular structure and properties of SARS-CoV-2 Mpro inhibitors (Appendix A). In detail, the ATS descriptor depicts the distribution of atomic properties (atomic masses, polarizability, charge and electronegativity) along with the topological structure of the molecule. Polarizability properties are also described by the Burden modified eigenvalues descriptors. Barysz matrix topological descriptors account for the presence of heteroatoms and multiple bonds; finally, CrippenLogP reports hydrophobicity properties.

Based on these outcomes, it seemed that parameters related to charge distribution, polarizability and electronegativity were crucial for the discrimination of actives in the dataset.

For SVM hyperparameters of C and γ types, we selected 100 and 0.01 values, respectively, while the kernel function was the radial basis function (RBF) [25]. The use of kernel functions in SVM, also called “kernel trick”, helped us to map the training data into a higher dimensional space. This function turned out to be essential in our model having linear non separable data.

### 2.3. Structure-Based Insights

#### 2.3.1. PDBs Study and Docking Protocol Validation

In order to select the best protein structure for the validation of the docking protocol, an extensive PDBs study was conducted.

Firstly, we analysed 25 Mpro co-crystallized PDBs structures to detect the key residues crucial for the inhibitor–protein interaction. Of the 25 structures analysed, only five (5RF6, 5RGW, 6WCO, 5R82 and 6W79) satisfied our criteria (see Section 3). On these 5 PDBs, the B-factor (PDB B-value Mean) was checked to assess the protein structure quality [26]. All the structures analysed presented B-factor values in an acceptable range for further studies (see Table 5).

We observed that the noncovalent binding mode was stabilized by hydrogen bonds to the Gly 143 and Glu 166 NHs and to the aromatic ring of His 163; additionally, a π−π interaction was observed with His 41.

The docking protocol was validated through cognate docking calculation runs, which assessed the ability of the docking algorithms to reproduce the correct binding mode of the co-crystallized ligands. The validation consisted in removing the co-crystallized ligand and in re-docking it into the active site. The re-docked complexes were then superimposed onto the reference co-crystallized complex and the root-mean-square deviation (RMSD) was calculated. Results are shown in Table 5.

The best cognate docking results were observed for 5RGW, 5R82 and 6WCO PDBs with RMSD values below 2 Å (which is considered the RMSD cut-off to assess docking accuracy). Despite the high docking accuracy, 5R82 PDB was excluded from further analysis, having a fragment-size co-crystallized ligand, while the larger and better fitted co-crystallized ligands of 6WCO and 5RGW were taken further. The binding poses of docked and crystallographic ligands are depicted in Figure 2.

#### 2.3.2. Molecular Dynamic Simulation

In order to verify the stability of the retrieved interactions within the crystal structure and discover new putative ones, 200 ns MD simulations on the two best performing PDBs (6WCO and 5RGW) were carried out. As seen from the RMSD and RMSF plots (Figure 3), during the whole 6WCO MD trajectory, the protein and the protein–ligand complex maintained a good stability. Moreover, stable interactions with the known crucial residues were observed during the MD (Appendix A). The simulation of 5RGW showed instead a less stable behaviour of the complex, compared to 6WCO (MD analysis of 5RGW is reported in Appendix A).

#### 2.3.3. Virtual Screening of Commercially Available Libraries

The final SVM model was applied for a preliminary screening of about 2 million compounds from commercial libraries (MolPort, Asinex and ChEMBL). Two hundred compounds were classified by the model as actives. On this reduced dataset, ADME parameters were calculated using Qikprop to filter only those presenting a safe predicted profile (see methods). Compounds that met ADME criteria were subsequently docked [27] and their binding mode was analysed. Compounds were prioritised based on the docking score and visual inspection.

The first five binding modes prioritized by the docking studies on the two PDBs were analysed and the retrieved interactions crucial for the binding mode were evaluated (Appendix A). Table 6 shows the interactions found by the docking runs. In Table 7, the five compounds binding mode in 2D and 3D are depicted.

Hydrogen bonds were the main non-covalent interactions involved in the predicted binding between ligands and the receptor and mostly involved residues, such as Gly143, His163, Glu166 and His41, according to the interaction performed by the PDB analysis and MD.

Of note, the sulfonamide moiety was recurrent in the top ranked compounds, suggesting a potential role of this moiety in the Mpro inhibitors. These results are indeed supported by the evidence that a large number of sulfonamide derivatives were reported to show antiviral activity [28].

Compounds with the most interesting binding poses according to the literature [3] were selected and will be biologically assayed against the viral protease.

#### 2.3.4. Molecular Dynamic Simulation Analysis

Based on the binding mode retrieved from the previous docking study, the consensus top ranked compounds were subjected to MD (100 ns), aiming at determining the stability of the protein–ligand complexes. RMSD values calculated for all frames in the trajectories revealed the stability of the protein conformation during the entire simulations. Figure 4 summarizes the interactions revealed by the five MD simulation runs.

From this analysis we observed that the interactions spotted by docking calculations were maintained as stable during the MD simulations. Moreover, new interactions emerged. In particular, Glu 166 had the highest interaction rate and was able to establish H-bond interactions with the ligands throughout the entire dynamic simulations. This residue is found as conserved in other coronaviruses [24]. This is of special relevance, because it has been reported that Glu166 is important for the protomer dimerization and catalytic activity of the protease [29,30,31].

Of note, some water-mediated H-bonds (His 41, Ala 46, Met 49, Asn142, Glu 166, Gln 189 and Thr 190) were also involved in the ligand protein interaction network.

Compounds III and IV experienced adjustments at the binding pocket, resulting in RMSD fluctuations. In particular, the isopropyl moiety of compound III and the nitrile group of compounds IV showed high rotamers mobility. The nitrile moiety of compound V maintained H-bond interaction with Gln 192 even during movements.

## 3. Material and Methods

### 3.1. Data Curation

PostEra COVID-19 Moonshot public database contains about 719 compounds and their reported activities are related to a fluorescence assay, by RapidFire mass spectrometry technology. The activity is expressed as the half inhibitory concentration (IC_50_) [32]. Activity data lead to the identification of the most and less potent compounds. Compounds were represented as SMILES strings, which were then converted into SDF format using the chemoinformatic tool rdkit [33]. In detail, SMILES strings were first converted in a Mol file; hydrogen atoms were added and, for each compound, a few conformations were generated using the ETKDG method [34]. With the SDF file as the input, the PaDEL software [35] calculated a total of 1444 1D and 2D type molecular descriptors. For each compound, the IC_50_ values were set as the labels. In order to select the most informative descriptors, no missing values were detected, while descriptors with zero variance were excluded from the dataset. Moreover, a correlation matrix was computed and high correlated features were dropped. This dataset cleaning process afforded a reduced number of 78 molecular descriptors.

The inactive compounds, with an IC_50_ higher than 98 µM, were excluded from the dataset, reducing the chance of introducing bias in the analysis.

The final dataset was randomly split into a training set (80%) and a test set (20%). The training set was standardized and the same scaling was applied to the test data, which were solely used during the evaluation stage. Standardization was performed using the Sklearn Standard Scaler class. Training set bioactivity values were discretized through the KBinsDisretizer class from Scikit learn library.

After training-set standardization, discretization technique was performed in order to transform the numerical input variables into discrete ordinal labels that led to the development of our machine learning model.

Continuous values of the training set were grouped into k = 2 discrete bins using the uniform method, making the data discrete. In this way, data were labelled in two categories, active and inactive, respectively, according to compounds corresponding IC_50_ values.

### 3.2. Feature Selection

Feature selection was performed by applying the RF method combined with the RF–RFE–CV methods on the training set (Figure 5). The RF–RFE–CV method was implemented by using Sklearn RFE–CV class, where random forest was set as the estimator.

Firstly, Sklearn random forest was performed in order to get information about the feature importance. Molecular descriptors significance was detected on the basis of their correlation with biological activity. At this point, it was necessary to set the number of decisional trees, being an important parameter for the forest population. We evaluated the model performance by setting a population of 1, 10, 100 and 500 trees [36].

Feature importance was ranked by performing a recursive feature elimination and a cross-validation, affording the best feature number selection. In particular, for each iteration, one feature was deleted at a time, until no further features were left to be removed. For the RFE–CV implementation, we defined a function using random forest as an estimator and setting the minimum number of features as one. This function returned the collection of the most informative molecular descriptors. Moreover, the RFE–CV applied a 5-fold cross validation method [38] (Figure 6).

### 3.3. Support Vector Machine

With the selected molecular descriptors in hands, we trained an SVM aiming at predicting the activity of novel Mpro inhibitors.

The SVM model was implemented in python 3 using Sklearn libraries. The SVM model was trained using the training set (80% of the data). Sklearn SVM class takes several parameters, such as kernel function, regulation parameter (C) and gamma parameter (γ).

SVM hyperparameter tuning was performed through a grid algorithm using Sklearn GridSearchCV. The specified grid hyperparameters set were the kernel parameter (RBF, poly and linear), C values (in a range between 1 × 10^0.001^ and 1 × 100^0.001^) and the γ parameter (range between 1.0 and 1 × 10^−3^). Next, the model was trained using the best SVM hyperparameters in terms of accuracy and precision, through the fit method, according to the given training data (Figure 7).

### 3.4. Structure-Based Approach

#### 3.4.1. Proteins and Ligands Preparation

Proteins were prepared using the Protein Preparation Wizard tool (Schrödinger, LLC) [39] in order to optimize their improprieties, such as missing hydrogens and missing loops, and to avoid atomic clashes. The protonation state was set in the pH range of 7.0 ± 2.0. Protein crystal structures were further optimized using energy minimization with the OPLS3e force field [40,41]. The receptor grid was centred on the co-crystallized ligand and the receptor Van der Waals radii was unscaled. Ligands were prepared using the Schrödinger LigPrep tool v. 2018-2 [39]. OPLS3e was again adopted as the force field (ff) and Epik was selected at a pH of 7.0 ± 2.0, as the ionization tool.

#### 3.4.2. PDB Study

From the PDB database [42], 25 structures containing co-crystallised ligands with a resolution between 1.0 and 1.5 Å (optimal range for a reliable interaction study) were obtained. The selected PDBs were analysed to verify that ligands bound non covalently to the catalytic site with known interactions. In Table 8, the identified PDB codes are reported.

Out of these 25 structures, only five (5RGW, 5R82, 6WCO, 5RF6, 6W79) have a co-crystallized ligand within the catalytic cavity. On these structures, electron density maps (2Fo-Fc) and B-values were analysed to assess that the interacting ligands were well covered and the overall structure quality. The analysis revealed a good fit on the electron density maps and reasonable B-values.

#### 3.4.3. ADME Filter and Docking Calculations

Compounds selected by the SVM model were filtered according to ADME criteria (Table 9).

The filtered compounds were docked using the Glide software (Schrodinger, L.LC) on 5RGW and 6WCO. The retrieved binding mode of the consensus prioritized molecules was analysed.

A maximum of 10 generated conformers was set. The binding site was defined using the co-crystallized ligands coordinates. Finally, 200 selected compounds from the commercial libraries were docked in standard precision mode (SP) and the top ranked poses were analysed [43].

#### 3.4.4. Molecular Dynamics

Molecular dynamic simulation (MD) was performed using the Desmond simulation package by Schrödinger LLC, v5.6 [44,45], on the co-crystallized Mpro crystal structure (PDBs 6WCO and 5RGW).

The key residues involved in ligand–protein complex stabilization were analysed by MD 200 nanosecond (200 ns) long, using a 0.002 ps (2.0 fs) time step. The complex was enclosed in an orthorhombic box and a TIP3P water model was used. The box volume was minimized and OPLS3e force field (ff) was applied. The same ff was used to perform the MD simulation. The simulation was performed at 300 K in an NPT ensemble. A Nosé–Hoover chain thermostat was used with a relaxation time of 1 ps. A Martyna–Tuckerman–Klein barostat was set to regulate the pressure with isotropic coupling and relaxation time of 2.0 ps. The complex stability evaluation between the putative Mpro inhibitors identified by consensus docking was performed by running MD simulations 100 ns long, under the same conditions reported above.

## 4. Conclusions

In this study, an SVM model was built for the prediction of inhibitory activity of novel chemo-types against SARS-CoV-2 Mpro. The model was implemented in python3 language using Sklearn libraries and was developed using PostEra COVID-19 Moonshoot public activity data. The main relevant molecular descriptors were selected through a random forest approach combined with a recursive feature elimination and a cross validation method (RF–RFE–CV). The final model was tested and showed an accuracy of 0.88. Finally, the model was used for the prediction of the inhibitory activity of compounds commercially available against the viral protease. These compounds were docked and the key residues for crucial interactions were retrieved, analysing the binding poses of ligand-protein co-crystallized complexes. Moreover, a deep binding study was carried on by performing MD simulations, which showed an acceptable complex stability for all the compounds analysed. Of high interest was the interaction of the best five ligands with Glu 166 of the protein. This residue, found as conserved in other coronaviruses, was demonstrated to be crucial in the dimerization of the Mpro protomers, that is the key event related to the catalytic activity of Mpro.

Compounds with the best binding poses will be evaluated in the biological primary assay and validated as promising Mpro inhibitors.

Of note, although the SVM model was built over a limited number of compounds, it turned out to be a valid approach for the identification of new potential SARS-CoV-2 Mpro inhibitors.

## Figures and Tables

**Figure 1 ijms-22-07714-f001:**
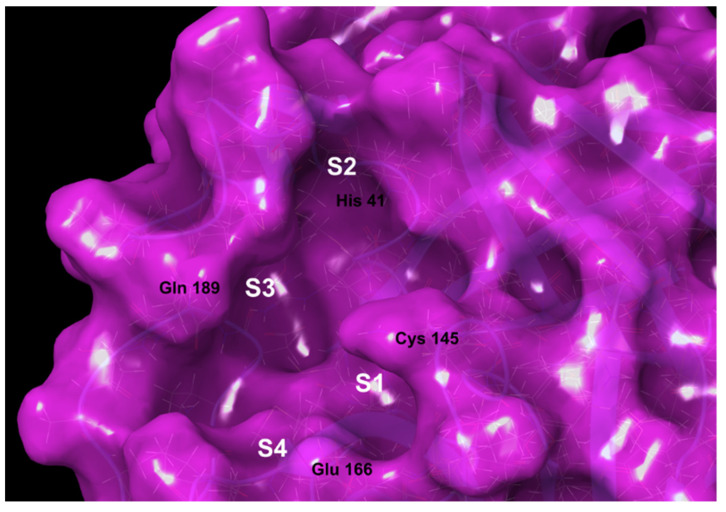
Focus on Mpro binding pocket (PDBid 6WCO). In white are highlighted the S1, S2, S3, S4 substrate binding pockets.

**Figure 2 ijms-22-07714-f002:**
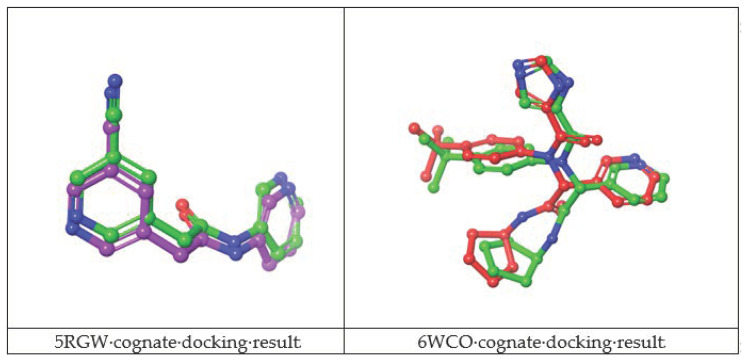
Superimposition of the docked 5RGW (purple), 6WCO (orange) and the crystallographic (green) conformations of the cognate ligands binding pose.

**Figure 3 ijms-22-07714-f003:**
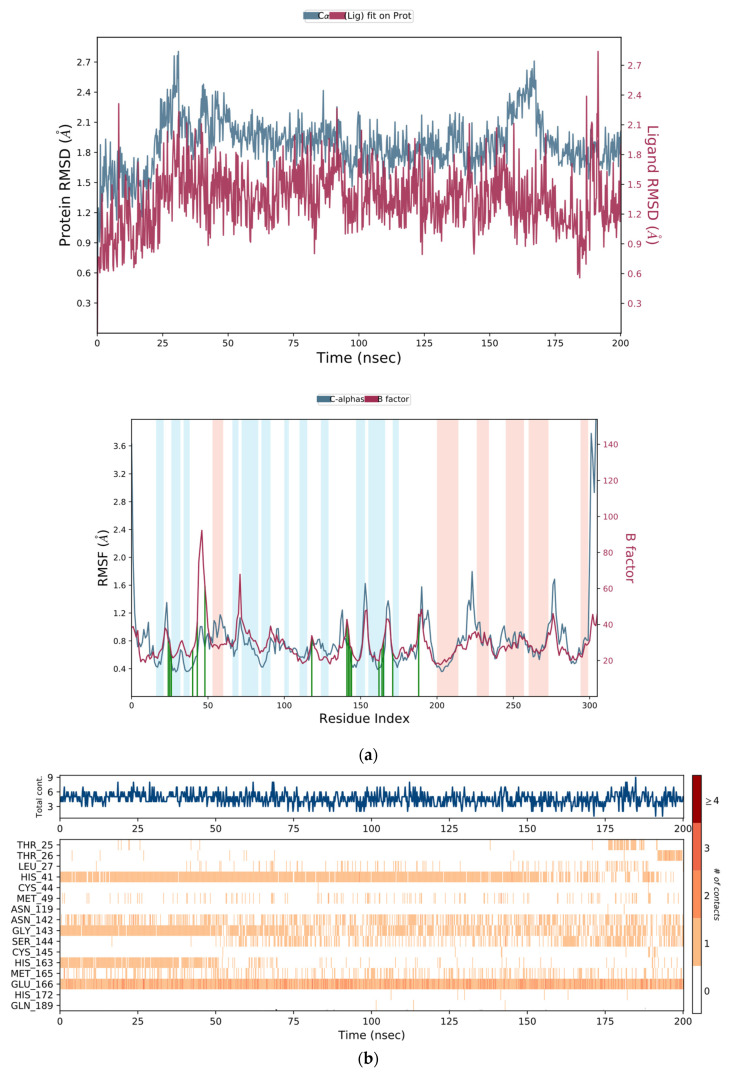
(**a**) 6WCO Molecular dynamic root-mean-square deviation (RMSD) and root-mean-square fluctuation (RMSF) plots. For the RMSF plot: ligand contacts are reported in green; orange zones represent α-helices, blue zones refer to β-sheets and white to not folded regions of the protein. (**b**) Protein–ligand contact timeline. Dark orange shades correspond to higher number of contacts.

**Figure 4 ijms-22-07714-f004:**
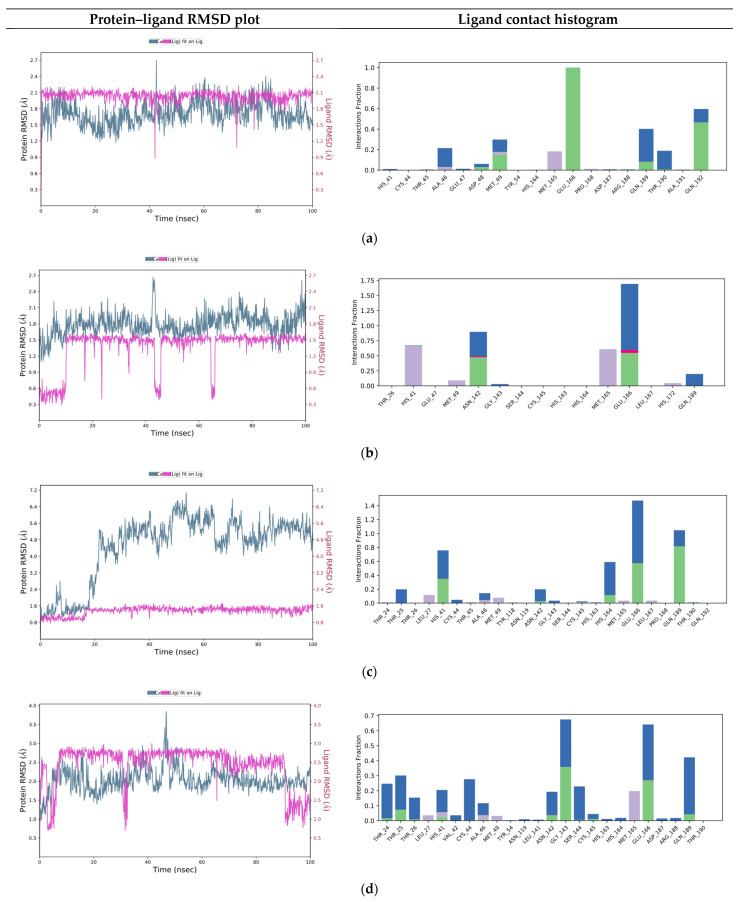
Five consensus top ranked compounds RMSD plots and protein ligand contacts histograms monitored throughout the MD simulations analysis. Green bar, H-bonds; blue bar, water mediated H-bonds; purple bar, hydrophobic. (**a**) Cmpd I. (**b**) Cmpd II. (**c**) Cmpd III. (**d**) Cmpd IV. (**e**) Cmpd V.

**Figure 5 ijms-22-07714-f005:**
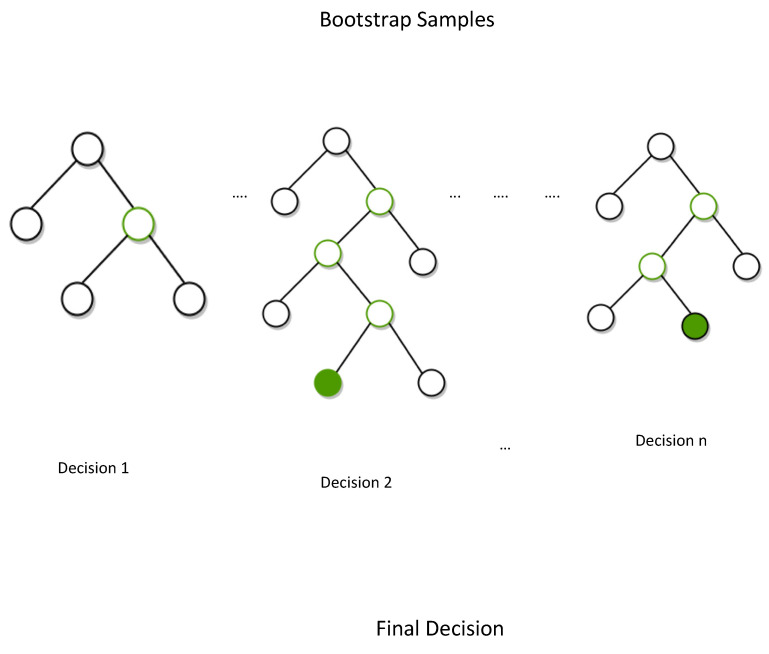
RF procedure. Each tree is built over a bootstrap sample (about 2/3 of the samples) of data and is used as a training set, in order to predict the data in the remaining 1/3, which is instead used as a test set sample (out-of-bag samples, or OOB) [17,36]. When a decision is made, the best predictor is identified and split on until the final decision is reached [37].

**Figure 6 ijms-22-07714-f006:**
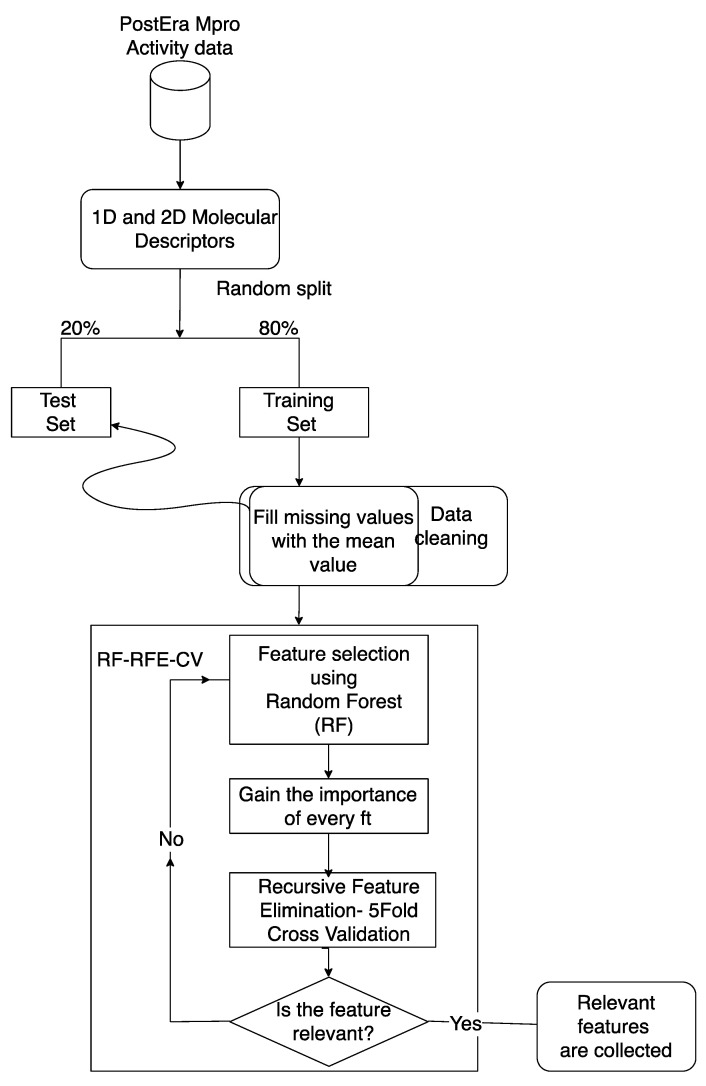
RF–RFE–CV data pre-processing and feature selection flowchart.

**Figure 7 ijms-22-07714-f007:**
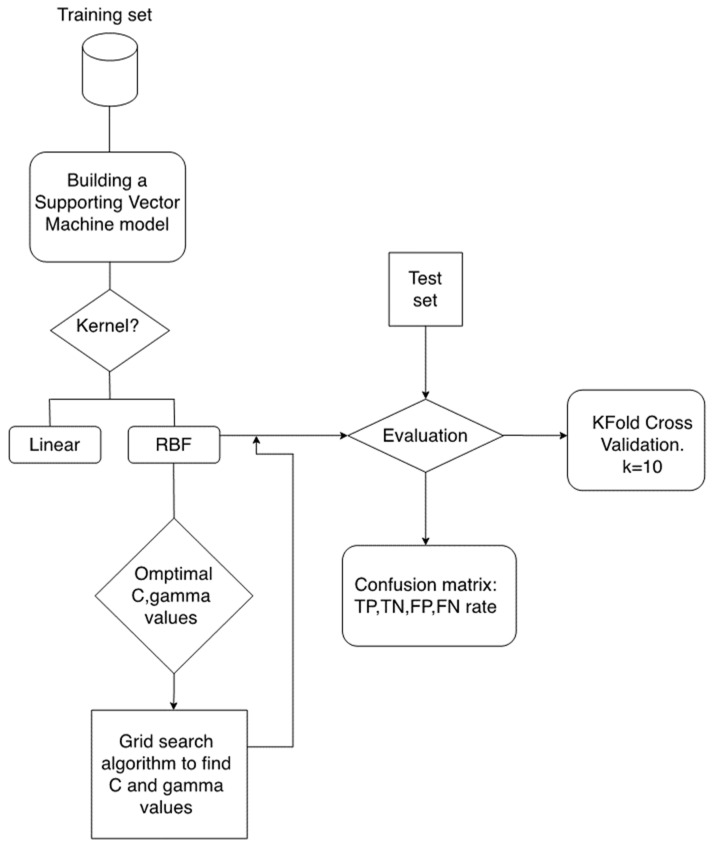
Flowchart of the building of the SVM model.

**Table 1 ijms-22-07714-t001:** Molecular descriptors distribution.

Molecular Descriptor Type	*n* Trees			
	1	10	100	500
Atom type electro-topological state	16			
Autocorrelation	12	24	2	6
Topological charge	7			
Barysz matrix	6	5	1	2
Ring count	3			
Extended topochemical atom	3			
Carbon types	3			
Molecular distance edge	3			
Burden modified eigenvalues	2	5	3	3
Detour matrix	2			
Chi chain	2			
Bcut	2	2		
Basic group count	1			
Molecular linear free energy relation	1			
Chi path	1			
Longest aliphatic chain	1			
Largest pi system	1			
Molecular linear free energy relation	1			
Petitjean number	1			
Crippen logP and MR		1	1	1
XlogP	1			
Number of selected descriptors	68	37	7	13

**Table 2 ijms-22-07714-t002:** Best SVM hyperparameter found by the grid search algorithm when 1, 10, 100 and 500 trees were set.

Support Vector Machine Best Hyperparameters			
*n* Trees	Kernel	C	γ
1	RBF *	10.0	0.1
10	RBF	1.0	0.01
100	RBF	1.0	1.0
500	RBF	100.0	0.01

* The radial basis function (RBF) kernel, in the event of non-linear separation, leads to map the data into a higher-dimensional space.

**Table 3 ijms-22-07714-t003:** Best correlation, accuracy and precision values obtained setting 1, 10, 100 and 500 decisional trees in RF–RFE–CV process.

*n* Tree	Correlation	Accuracy	Precision
1	0.65	0.85	0.66
10	0.74	0.84	0.75
**100**	**0.75**	**0.88**	**0.75**
500	0.90	0.83	0.5

**Table 4 ijms-22-07714-t004:** Feature selection analysis of the selected molecular descriptors of the best classification model developed. Permuted feature importance score is reported with the standard deviation.

Optimal *n* Features Selected	Descriptor	Descriptor Type	Class	Permuted Feature Importance
7	AATS6i	Autocorrelation	2D	0.044 ± 0.006
	ATSC7m	Autocorrelation	2D	0.047 ± 0.004
	VE1_DzZ	Barysz matrix	2D	0.086 ± 0.009
	SpMax2_Bhm	Burden modified eigenvalues	2D	0.070 ± 0.002
	SpMax1_Bhv	Burden modified eigenvalues	2D	0.052 ± 0.005
	SpMax2_Bhv	Burden modified eigenvalues	2D	0.076 ± 0.006
	CrippenLogP	Crippen logP and MR	2D	0.074 ± 0.005

**Table 5 ijms-22-07714-t005:** List of the best 5 PDBs with B-value and cognate docking results RMSD, expressed in (Å).

PDB	B-Value Mean (Å)	RMSD (Å)
5RGW	27.44	0.50
5R82	18.40	0.78
6WCO	36.09	1.11
5RF6	20.45	3.34
6W79	30.07	4.5

**Table 6 ijms-22-07714-t006:** Residue analysis of the interaction between putative Mpro inhibitors with 6WCO and 5RGW.

		6WCO		5RGW	
Cmpd	AMINO Acid	Protein Atom	Interaction	Protein Atom	Interaction
I	Gly143			NH	H-bond
	Glu166	CO	H-bond	NH	H-bond
	Thr190	NH	H-bond		
	Gln192	NH	H-bond *^a^*		
II	Gly143	NH	H-bond	NH	H-bond
	His163	NH	H-bond		
	His41	NH	H-bond	Ar *^b^*	π−π
	Ser 144	NH	H-bond		
III	Glu166	NH	H-bond	NH	H-bond
IV	Gly143	NH	H-bond	NH	H-bond
	His163	NH	H-bond *^a^*		
	Glu166			NH	H-bond
V	Gly143			NH	H-bond
	His41	Ar *^b^*	π−π		
	His163			NH	H-bond
	Thr190	NH	H-bond		
	Gln192	NH	H-bond		
	Glu 166	NH	H-bond		

*^a^* Side chain interaction; *^b^* aromatic ring.

**Table 7 ijms-22-07714-t007:** 2D structure and binding mode of the five consensus prioritized compounds versus the viral Mpro retrieved by docking calculations. Purple arrows represent H-bond interactions, blue lines surrounding the molecules underline non-polar regions and orange lines highlight negative charge residues, whereas green lines point polar residues.

Cmpd	Docking Pose	Ligand Interaction
I	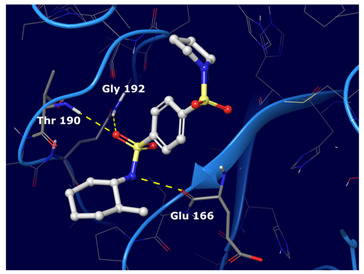	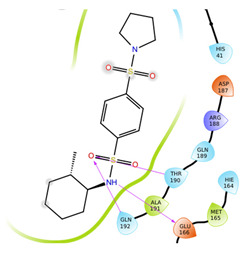
II	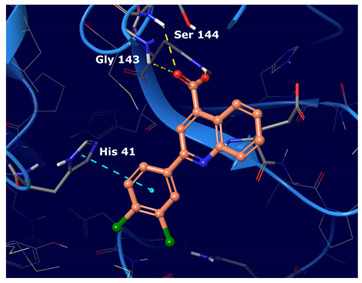	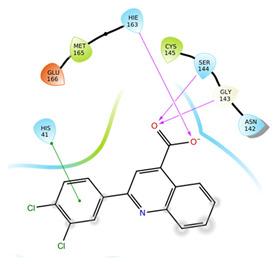
III	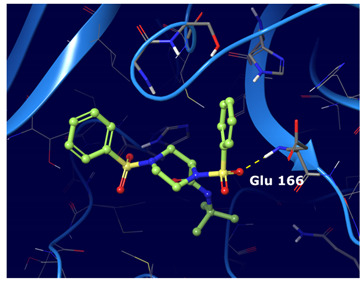	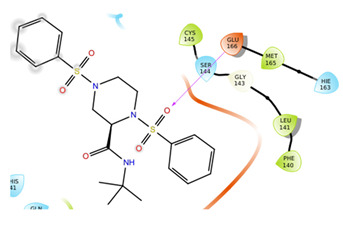
IV	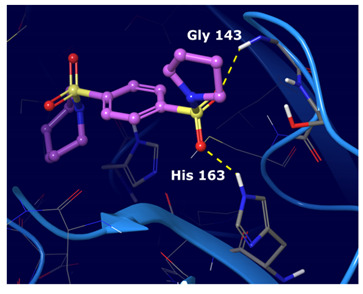	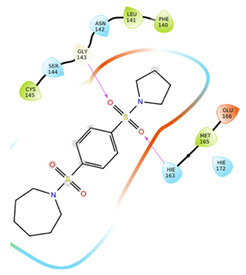
V	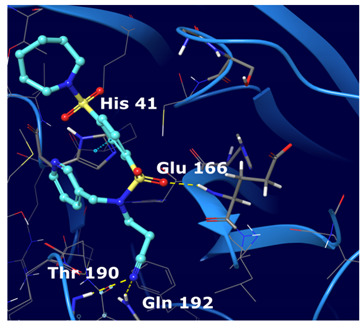	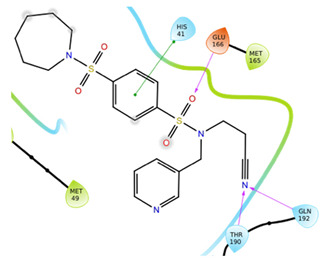

**Table 8 ijms-22-07714-t008:** PDB codes of Mpro with co-crystallized ligand having resolution < 1.5 Å.

PDB Codes				
5RGW	7AWR	5RGJ	5RFW	5RGR
5R82	7D1M	7AQE	5RH4	5RGK
6WCO	5RF9	7AOL	5RFC	5RED
5RF6	7K6D	6W79	5FRV	6XR3
6W79	7K40	5RL2	7AXM	5RF8

**Table 9 ijms-22-07714-t009:** ADME criteria.

Propriety	Description	Range of Values
QPlogS	Predicted aqueous solubility	7–200
QPlogHERG	Predicted IC_50_ value for blockage of HERG K^+^ channels	<−5
QPlogPo/w	Predicted octanol/water partition coefficiency	−2–6.5
QPPCaco	Predicted apparent Caco-2 cell permeability, in nm/sec	>25
Rule of five	Number of violations of Lipinski’s rule of five	≤3
mol_MW	Molecular weight of the molecule	>250
# rotor	Number of non-trivial and non-hindered rotatable bonds	<10

## Data Availability

Data source: https://covid.postera.ai/covid/activity_data (accessed on 27 May 2019).

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
