# Peer review of "Support Vector Machine as a Supervised Learning for the Prioritization of Novel Potential SARS-CoV-2 Main Protease Inhibitors"

_ijms, 2021, doi:10.3390/ijms22147714_

Round 1

Reviewer 1 Report

I am happy with the way the paper was improved (especially that previously I recommended to reject the manuscript). The authors did a really good job, congratulations!

One thing I would like to ask is to adjust the Y-axis scale of the RMSD plots at Figure 4 to have the same range (well in this case that should be 7.2 for the protein and 7.2 for the ligand), as well as the same for the Interactions Fraction plots.

Author Response

Answer to Reviewer 1

Comments and Suggestions for Authors

I am happy with the way the paper was improved (especially that previously I recommended to reject the manuscript). The authors did a really good job, congratulations!

We thank Reviewer 1 for the positive feedback. We are very happy that the manuscript was improved and obtained reviewer approval.

One thing I would like to ask is to adjust the Y-axis scale of the RMSD plots at Figure 4 to have the same range (well in this case that should be 7.2 for the protein and 7.2 for the ligand), as well as the same for the Interactions Fraction plots.

We thank reviewer 1 for the suggestion. We adjusted the Y-axis scale of the RMSD plots in figure 4 (please see line 281). Unfortunately, we could not do the same for the “interaction fraction plots” since the tool used for generating it did not allow us; in fact, each diagram is generated individually. We are sorry and hope that the referee understands.

Reviewer 2 Report

In the “ijms-1304901-peer-review-v1.pdf” manuscript, the authors have made substantial research efforts to come up with a viable theoretical solution that might be useful to combat the most widespread global disease, COVID-19, by proposing a supervised learning approach for the prioritization of novel potential SARS-CoV-2 main protease inhibitors, followed by molecular docking, and molecular dynamics simulation.  Following this approach, five novel potential SARS-CoV-2 Mpro inhibitors have been detected.

The significant outcomes provided by the manuscript could be a real win for researchers interested in developing new antiviral agents potentially able to combat the current pandemic and beyond. Furthermore, in the case of positive results from clinical trials with COVID-19 patients, the identified compounds may pave the way for new antiviral agents potentially capable to fight the disease.

Considering the potential impact of the manuscript results in the research world, and with all the respect for the author's impressive work, the manuscript should be accepted for publication in the International Journal of Molecular Sciences, after minor revision.

Please find enclosed five minor questions addressed to the manuscript authors:

  1. In Figure 1, the Mpro binding pocket of 6Y2F is presented. This PDB structure is not part of the present study. Why did the authors choose to present this structure instead of the structures (5RGW, 6WCO) that were chosen for the molecular docking/molecular dynamics procedure? Please replace 6Y2F from Figure 1 with a relevant structure for this study.
  2. A short description of the four subsites (S1, S2, S3, and S4), the presentation of the amino acids in each substrate binding pocket is required in the main text of the manuscript. The appropriate reference must be inserted.
  3. Please check line 378: “…extra precision mode (SP) …”
  4. The "descriptor significance” for the most relevant selected molecular descriptors presented in Table 4 is missing.
  5. In addition to resolution, the B - factor is very important as a criterion in choosing PDB structure. Did the authors check the values of this factor? Please complete.

Author Response

Answer to Reviewer 2

Comments and Suggestions for Authors

In the “ijms-1304901-peer-review-v1.pdf” manuscript, the authors have made substantial research efforts to come up with a viable theoretical solution that might be useful to combat the most widespread global disease, COVID-19, by proposing a supervised learning approach for the prioritization of novel potential SARS-CoV-2 main protease inhibitors, followed by molecular docking, and molecular dynamics simulation.  Following this approach, five novel potential SARS-CoV-2 Mpro inhibitors have been detected.

The significant outcomes provided by the manuscript could be a real win for researchers interested in developing new antiviral agents potentially able to combat the current pandemic and beyond. Furthermore, in the case of positive results from clinical trials with COVID-19 patients, the identified compounds may pave the way for new antiviral agents potentially capable to fight the disease.

Considering the potential impact of the manuscript results in the research world, and with all the respect for the author's impressive work, the manuscript should be accepted for publication in the International Journal of Molecular Sciences, after minor revision.

Please find enclosed five minor questions addressed to the manuscript authors:

  1. In Figure 1, the Mpro binding pocket of 6Y2F is presented. This PDB structure is not part of the present study. Why did the authors choose to present this structure instead of the structures (5RGW, 6WCO) that were chosen for the molecular docking/molecular dynamics procedure? Please replace 6Y2F from Figure 1 with a relevant structure for this study.

We thank the reviewer for bringing this point up. We have changed Figure 1 according to the suggestion and used 6WCO PDB structure for the image.

  1. A short description of the four subsites (S1, S2, S3, and S4), the presentation of the amino acids in each substrate binding pocket is required in the main text of the manuscript. The appropriate reference must be inserted.

We thank the reviewer for this suggestion. In the current version of the manuscript, we added a short description of each substrate binding pocket with the appropriate references. Please see lines 46-53.

  1. Please check line 378: “…extra precision mode (SP) …”

Thank you for pointing it out. We revised the sentence accordingly.

  1. The "descriptor significance” for the most relevant selected molecular descriptors presented in Table 4 is missing.

Descriptor significance for the most relevant selected molecular descriptors presented was added in table 4. Please see line 172.

  1. In addition to resolution, the B - factor is very important as a criterion in choosing PDB structure. Did the authors check the values of this factor? Please complete.

We thank the reviewer for the very good advice. We revised the manuscript accordingly. Please see lines 195-199 and Table 5 line 212.

Reviewer 3 Report

1- abstract have to be changed to represent the current work not to introduce the work.

Author Response

Answer to Reviewer 3

Comments and Suggestions for Authors

  1. abstract have to be changed to represent the current work not to introduce the work.

We thank the Reviewer for this important observation. We implemented the abstract accordingly (please see lines 19-26).